# Operationalizing Ethics for AI Agents:
# How Developers Encode Values into Repository Context Files

## Abstract

As AI coding agents become embedded in software development workflows, developers are beginning to operationalize ethical principles by encoding behavioral rules into repository-level context files, such as `AGENTS.md` files. Rather than examining the ethics *of* AI agents in the abstract, this vision paper investigates how ethics and values are already being translated for AI agents into actionable instructions that shape agent behavior. Through a preliminary investigation, we find that developers are already embedding guidance related to fairness, accessibility, sustainability, tone, and privacy. These artifacts function as a developer-authored governance layer, translating abstract principles into situated, natural-language directives within development workflows. In this **position paper**, we outline a research agenda for studying this emerging practice, including how encoded values vary across communities, what governance dynamics emerge when multiple contributors negotiate these files, and whether agents reliably adhere to the constraints specified. Understanding how ethics and values are operationalized for AI agents is essential to ground AI governance in modern software engineering practice.

**ACM Reference Format:**
Anonymous Author(s). 2026. Operationalizing Ethics for AI Agents: How Developers Encode Values into Repository Context Files. In *Proceedings of 3rd ACM International Conference on AI-powered Software (AIware '26)*. ACM, New York, NY, USA, 3 pages. https://doi.org/XXXXXXX.XXXXXXX

## 1 Introduction

Ethical principles for AI systems, such as fairness, accountability, transparency, and safety, are widely discussed in research and policy. However, translating these principles into everyday engineering practice remains difficult. Principles might be too vague, unenforceable, or downright clash with each other [5]. Engineers may struggle to translate ethical principles into actionable design requirements.

The increasing integration of AI coding agents into development workflows introduces a new dimension to this long-standing challenge. These agents—with varying degrees of autonomy and human involvement—generate code, propose refactorings, review pull requests, and interact directly with repository artifacts. Their behavior is shaped not only by model training and prompts, but also by contextual artifacts embedded in repositories.

In this paper, we argue that a new and largely unexplored phenomenon is emerging: developers are operationalizing ethics *for* AI agents by encoding behavioral constraints directly into repository-level context files. Files such as `AGENTS.md` contain instructions that specify how AI agents should behave within a project. These instructions are written by humans, interpreted by machines, and embedded in development workflows.

Rather than treating AI ethics as an overarching, external governance mechanism, developers are embedding ethical commitments directly into their projects. Taking into account contemporary issues in AI and digital ethics, this translation layer between abstract principles and concrete agent behavior constitutes a new object of study for software engineering research.

## 2 Background and Related Work

### 2.1 Ethics, values, and SE

Ethics and values have long been discussed in software engineering and in digital ethics, *writ large*. Foundational efforts such as the *ACM/IEEE Software Engineering Code of Ethics* articulate professional obligations and societal responsibilities of software engineers [9], while various other frameworks and principles articulate the same on a broader level [5, 10]. However, several authors argue that such codes provide limited actionable guidance for everyday design decisions and must be complemented by explicit ethical deliberation within development teams [8, 15], resonant with critiques of applied AI/digital/technological ethics "...in general... lack[ing] mechanisms to reinforce its own normative claims" [10].

Beyond professional conduct and on a broad practice- and policy-level, research has explored how ethical values can be integrated into software processes. Value-Sensitive Design [6] and Value-Based Engineering [20] aim to make human values explicit in system design, while the IEEE 7000 standard proposes structured process models to address ethical concerns during system development [19]. Systematic reviews show growing interest in operationalizing human values and acceptable norms [7] in software engineering, particularly in the requirements and design phases, but highlight limited support for later lifecycle stages and implementation practices [2, 18].

Several approaches translate ethical principles into concrete development artifacts. *Ethical User Stories* and related agile practices embed concerns such as fairness, accessibility, and sustainability into backlog items and sprint routines [11, 22]. Goal-oriented methods derive *Social, Legal, Ethical, Empathetic, and Cultural* (SLEEC) requirements from explicit value models [12], while recent work advocates for lightweight, proactive integration of such considerations into existing engineering workflows [13].

### 2.2 Agents and Agents4SE

Studies on *in silico* agents, *viz.* the increasing need for ethical conduct in their use, design, deployment, and emergent behaviors, are

not new concepts. Consider traditional agent-based social simulations: ethical issues are found to "...arise from both its practice and its organisation" [3]. This 'top-down approach' (from the practitioner's perspective) is also complemented by a similar call from the 'bottom-up approach' (from the agent's perspective), such as the operationalization of 'good' versus 'bad' behavior at agentic level [16]. Needless to say, many of these studies pre-date modern Generative AI technologies.

In parallel, the rise of AI coding agents has spurred research on agent architectures, benchmarks, and human–AI collaboration. The SE community proposed trustworthiness frameworks, which incorporate principles such as fairness, robustness, and transparency (e.g., CRAFT values) for AI agents or *AI software engineers*. [1]. However, this work primarily focuses on agent capabilities and evaluation, rather than how developers configure persistent behavioral constraints within repositories. In other words, this concerns explicit guidance of norms and proper conduct of agents (*vis-à-vis* their deployers) from the 'bottom-up', as we argued previously.

Recent research has begun to conceptualize repository-level context files as a distinct class of artifacts in agentic software development. Large-scale analyses of such files ("Agent READMEs") like CLAUDE.md and AGENTS.md show that they resemble configuration files, primarily encoding operational guidance such as build instructions, implementation constraints, architectural information, and project-specific policies [4, 17]. Empirical evaluations further suggest that AGENTS.md can influence agent efficiency without degrading task completion, and their rapid adoption across tens of thousands of repositories indicates that such files are becoming an established component of agent-enabled workflows [14].

In summary, previous work has examined professional ethics, value-based development processes, agent trustworthiness, considerations for ethics in agent-based studies, and the structure and efficiency implications of agent context files. However, the specific question of *how* ethical principles are translated into machine-interpretable repository artifacts for AI agents remains largely unexplored. Our work addresses this gap by conceptualizing AGENTS.md files as a concrete operational layer for encoding ethics in agentic software development.

## 3 Preliminary Investigation

To explore how developers operationalize ethics for AI agents, we examined software repositories containing AGENTS.md files with explicit behavioral instructions using GitHub search. Even in this small exploratory sample, ethical principles are not merely referenced but translated into concrete, agent-directed constraints. Table 1 presents representative verbatim excerpts illustrating these different forms of ethical operationalization.

For example, one repository instructs the agent to "follow all guidelines for ethical AI", explicitly emphasizing keeping "the human in the loop", "taking accountability for changes", and being "transparent". While this appears to translate abstract notions of oversight into interaction-level constraints, it remains unclear what concrete behavioral changes an AI agent could derive from such high-level and ambiguous directives. The instruction documents ethical intent, but it does not specify executable conditions,

**Table 1: Examples of Ethical Operationalisation in Agent Context Files**

| Link | Verbatim Excerpt |
|---|---|
| Abhiek187/ez-recipes-web | "When working with the user, ensure you follow all guidelines for ethical AI, such as keeping the human in the loop, taking accountability for changes, and being transparent..." |
| github/awesome-copilot | "Build systems that are accessible, ethical, and fair. Test for bias..." [...] test_names = ['John Smith', 'José García', 'Lakshmi Patel', 'Ahmed Hassan', '李明'] [...] "Different outcomes for same qualifications but different names" |
| D7460N/DHCP | "Explicitly Avoid ...Moral lectures or unsolicited opinions" |
| haxtheweb/create | "Accessible: HAX maximizes accessibility..." [...] "Sustainable: Environmental (less data, lower battery usage)...and community...sustainability." |
| home-assistant/core | "Inclusivity: Use objective, non-discriminatory language" [...] "Clarity: Write for non-native English speakers" |
| tmobile/magentaA11y | "Respectful, Inclusive Language..." [...] "Bias-Aware and Error-Resistant..." [...] "Verification-Oriented Responses..." |

triggers, or enforcement mechanisms. In contrast, another repository operationalizes fairness through structured bias testing: beyond requiring systems to be "accessible, ethical, and fair", it provides explicit test data such as ['John Smith', 'José García', 'Lakshmi Patel', …] and directs the agent to check for "different outcomes for same qualifications but different names", embedding fairness as executable evaluation logic rather than a high-level aspiration. Other repositories encode normative boundaries on agent behavior, for instance by instructing agents to "avoid moral lectures or unsolicited opinions", or by mandating "objective, non-discriminatory language" and communication suitable for non-native English speakers. Across these examples, accountability, fairness, inclusivity, and tone are reformulated as machine-directed constraints that are directly intended to shape agent behavior.

These examples demonstrate that developers are actively selecting the ethical concerns that matter in their projects and reformulating them as machine-directed instructions. Fairness becomes bias testing logic, accountability becomes interaction constraints, inclusivity becomes linguistic guidance, and sustainability becomes a design requirement. Ethical commitments are therefore not simply declared at the level of principles; they are embedded directly into the instructions that are meant to shape agent behavior within development workflows. Although existing attempts are

laudable, this practice is still in its nascent stages and leads to further inquiry: given that we humans ourselves constantly grapple with 'doing the right thing' and 'doing things right', it remains unclear how AI agents can be expected to navigate these distinctions when guided only by natural-language repository directives.

## 4 Roadmap

The emergence of repository-level context files that encode behavioral constraints for AI agents has introduced a new dimension to discussions of ethics and AI: rather than debating the ethics of AI agents in the abstract, developers are beginning to operationalize ethics for AI agents within everyday development workflows. This development opens up a new research frontier for software engineering. A first step is large-scale empirical mapping of repository-level context files to identify which ethical categories are encoded most frequently and which remain absent. This work can reveal whether bias, accessibility, sustainability, or privacy dominate developer attention and how these emphases vary across domains and cultures. This will also reveal the value choices developers prioritize over others when it comes to instructing their agents: which, again, reflects on the diversity of values we humans uphold.

Beyond categorization, it is important to study the translation process itself. How do developers decide which ethical principles to encode? How broad (or specific) are these ethical directives in such files? How are these constraints negotiated in pull requests? How are these principles negotiated between human software engineers and AI counterparts? Mining repository histories combined with qualitative analysis can uncover the socio-technical dynamics underlying ethical operationalization.

Equally important is evaluating whether agents adhere to encoded constraints. Experiments can compare agent output with and without context files, measuring compliance with bias mitigation rules, tone requirements, accessibility standards, and regulatory compliance. Such studies would illuminate whether operationalization effectively shapes behavior or simply signals intent.

Longitudinal analysis can further examine how 'encoded ethics' evolves. Ethical commitments and/or directives can be strengthened after incidents, relaxed under productivity pressure, or adapted to new regulations. Some might be an exercise in 'boxticking', while others are genuine commitments to ethics, with varying degrees of practice, paralleling the trend of ethical documentation for *humans* [7]. Repository histories make it possible to study how such shifts unfold over time.

At the same time, it remains unclear whether how ethics are communicated to AI agents should resemble the way ethics are traditionally documented for human contributors, such as in codes of conduct or policy statements [21]. Human-oriented ethical documentation often relies on shared norms, contextual judgment, and implicit understanding. Determining whether similar forms of documentation suffice or whether fundamentally different, machine-oriented representations are required is an important direction for future research.

Developers are already operationalizing ethics for AI agents in today's repositories. By studying how these operational constraints are encoded, interpreted, and revised, software engineering research can move beyond abstract principles to an empirical understanding of how AI governance is implemented in practice. Moving forward, these insights can be applied beyond SE to encompass AI agent behavior in other domains, including business and social sciences.

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
