# OpenReview forum: "Operationalizing Ethics for AI Agents: How Developers Encode Values into Repository Context Files"
_ACM.org/AIWare/2026/Conference — AIware 2026_

### Official Review · Reviewer_M9kc · 2026-02-23

**Rating:** 4
**Confidence:** 4

**Review:**

Pros:

1. Quality: the investigation is very limited but it does show that developers have already started to encode values in the context files to guide and constrain agent behaviors.

2. Clarify: the presentation is clear.

3. Originality and significance of this work: the idea is novel and impactful. It convinces readers that this is a good research direction.

Cons:

The result would be much more convincing if the preliminary investigation includes some analysis on whether the example ethical operationalization in the context files actually changed the agents' behavior (and how).

**Summary:**

This position paper outlined a research agenda for investigating how developers embed guidance related to fairness, accessibility, sustainability, tone, and privacy into repository context files. Through a preliminary investigation, the authors claimed that developers have already started to encode values in the context files to guide and constrain agent behaviors.

---

> ### Author Response · Authors · 2026-03-21
>
> Thank you for your constructive review.
>
> We agree that evaluating whether encoded ethical constraints influence agent behavior is an important next step. For the camera-ready version, we will clarify in the roadmap section that future work should experimentally evaluate agent behavior with and without repository context files. Concretely, we will add a sentence along the lines of:
>
> “Future work should experimentally compare agent behavior with and without repository context files, for example by measuring compliance with constraints related to bias mitigation, tone, accessibility, or regulatory guidance.”
>
> Because this paper is intended as a position paper, we did not include such experiments here, but we will make this evaluation direction more explicit in the discussion of future research.

---

### Official Review · Reviewer_CCWz · 2026-03-10

**Rating:** 3
**Confidence:** 4

**Review:**

Method and Rigor. This paper discusses how developers encode ethical guidance for AI agents in repository context files such as AGENTS.md. The idea is interesting and the examples illustrate how ethical principles can be translated into instructions for agents. However, the empirical part of the paper is very small and mostly illustrative. The authors mention conducting a preliminary investigation using GitHub search, but the process is not described in detail. It is not clear how repositories were found, how many were examined, or how the examples in the table were selected. For a position paper this is understandable, but a short explanation of the search process would still strengthen the credibility of the examples and help readers understand the scope of the observation.

-----

Goal, Results, and Relevance. The goal of the paper is to argue that developers are starting to operationalize ethical principles for AI agents by encoding guidance directly into repository files. This is a relevant topic, especially with the increasing use of AI agents in development workflows. I see the main contribution of the paper as the idea that these files could represent a new layer where ethical considerations are embedded in development practice. The roadmap presented in the final section proposes several interesting research directions. However, the argument relies heavily on a small number of examples, so the claim that this is an emerging practice would benefit from slightly stronger grounding.

-----

Presentation. The paper is well written and easy to follow. I think that, other than some details about how the information was collected from the repositories, the paper has the necessary details for a position paper.

------

Overall Perception. The topic is interesting and relevant for the AIware community. As a position paper, the goal is mainly to raise a research agenda, which the paper does reasonably well.

**Summary:**

This is a position paper focused on how developers encode ethical guidance for AI coding agents in repository context files and on describing research directions to study this emerging practice in software development.

---

> ### Author Response · Authors · 2026-03-21
>
> Thank you for your constructive review.
>
> For the camera-ready version, we will include a description of the exploratory search process used to identify repositories with agent context files. Specifically, we will clarify that we used a combination of GitHub code search and ChatGPT web search to identify repositories containing AGENTS.md files with potential ethics-related guidance. We then manually inspected 25 repositories and selected examples, with six illustrative examples presented in Table 1.
>
> We will also refine the wording around the claim that this represents an emerging practice to better reflect that the current examples serve as illustrative observations motivating the research agenda, rather than a comprehensive empirical study.

---

### Official Review · Reviewer_tN7Y · 2026-03-11

**Rating:** 3
**Confidence:** 5

**Review:**

+ The core claim of repository-level artifacts providing an operational governance layer is relevant and logically developed.
+ The roadmap outlines empirical mapping and methodological exploration.

- Research Gap Positioning: There is prior work on operationalizing RAI through governance and engineering practices. For example, the RAI Pattern Catalogue [A] presents governance, process, and product patterns that translate ethical principles into system-level practices (e.g., standardized reporting, ethical requirements, governance structures, documentation templates). The current paper should clarify that how repository-level “contextual artifacts” differ from RAI governance/process/product patterns and why this layer has not already been covered by existing Responsible AI engineering frameworks.
[A] Q. Lu, L. Zhu, X. Xu, J. Whittle, D. Zowghi, and A. Jacquet, ‘Responsible AI Pattern Catalogue: A Collection of Best Practices for AI Governance and Engineering’, ACM Comput. Surv., vol. 56, no. 7, Apr. 2024.

- Framing AI Ethics as “External”: The claim (Line 72) that AI ethics is typically treated as external governance overlooks existing work on ethics-by-design, Embedded governance mechanisms, Responsible AI engineering processes and Documentation templates. A brief acknowledgment of these works, along with clarification of how repository-level agent-facing artifacts differ would improve conceptual novelty. For example, direct machine-interpretable constraints in configuration files are different from human-facing governance structures.

-  The preliminary repository analysis (Lines 144–163) lacks sufficient detail. Even for a position paper, minimal methodological transparency is required to assess credibility. The authors can specify: Number of repositories analyzed, Search query and if any filters used, Inclusion/exclusion criteria, how they defined agent-targeted artifacts and repository characteristics etc. A small summary table including these details will improve transparency.

- The preliminary repository analysis (Lines 144–163) lacks sufficient detail. Even for a position paper, minimal methodological transparency is required to assess credibility. The authors can specify: Number of repositories analyzed, Search query and if any filters used, Inclusion/exclusion criteria, how they defined agent-targeted artifacts and repository characteristics etc. A small summary table including these details will improve transparency.

**Summary:**

This position paper argues that ethical principles for AI systems should be operationalized through repository-level artifacts (e.g., AGENTS.md or similar context files) that establish constraints and expectations for AI agents interacting with software repositories. The paper mentions these artifacts as a practical governance layer bridging abstract Responsible AI (RAI) principles and executable development workflows. It provides initial observations of such artifacts in open-source repositories and outlines a research agenda around empirical mapping, agent negotiation, and evaluation of encoded ethics. The topic is relevant as the integration of autonomous AI agents in software engineering workflows is increasing.

---

> ### Author Response · Authors · 2026-03-21
>
> Thank you for your constructive review.
>
> For the camera-ready version, we will clarify the distinction between repository-level context files and existing Responsible AI governance and engineering approaches, and cite the suggested work on the Responsible AI Pattern Catalogue. Specifically, we will add a clarification in the related work section along the lines of:
>
> “While existing Responsible AI frameworks translate ethical principles into governance processes and documentation artifacts for human developers, repository context files such as AGENTS.md contain instructions intended for AI agents and are embedded directly in software repositories and interpreted by AI agents during development workflows.”
>
> We will also revise the sentence suggesting that AI ethics is typically treated as external governance to acknowledge existing work on ethics-by-design, embedded governance mechanisms, and Responsible AI engineering processes, and cite representative work in these areas. We will clarify that repository context files introduce a distinct operational layer, consisting of machine-interpretable constraints intended for AI agents and embedded directly in repositories, which differs from the primarily human-facing governance mechanisms discussed in prior work. We will also emphasise the nature of ethics being a dynamic, nuanced area: in that, even in human endeavors, ethical frameworks and other mechanisms are good guides, but not a panacea; as such, AI agents will also have similar conundra.
>
> We will add methodological details for the preliminary investigation. Specifically, we will clarify that we used a combination of GitHub code search and ChatGPT web search to identify repositories containing AGENTS.md files with potential ethics-related guidance. We then manually inspected 25 repositories in more detail and selected examples, with six illustrative examples shown in Table 1.